# Neuro2Semantic: A Transfer Learning Framework for Semantic Reconstruction of Continuous Language from Human Intracranial EEG

## Abstract

Decoding continuous language from neural signals remains a significant challenge in the intersection of neuroscience and artificial intelligence. We introduce Neuro2Semantic, a novel framework that reconstructs the semantic content of perceived speech from intracranial EEG (iEEG) recordings. Our approach consists of two phases: first, an LSTM-based adapter aligns neural signals with pre-trained text embeddings; second, a corrector module generates continuous, natural text directly from these aligned embeddings. This flexible method overcomes the limitations of previous decoding approaches and enables unconstrained text generation. Neuro2Semantic achieves remarkable performance with as little as 30 minutes of neural data, significantly outperforming a recent state-of-the-art method in low-data settings. These results highlight the potential for practical applications in brain-computer interfaces and neural decoding technologies.

## 1 Introduction

Recent advances at the intersection of artificial intelligence (AI) and neuroscience have enabled powerful new modeling capabilities, particularly in the development of neural decoding models. These models aim to reconstruct stimuli or intentions based on measured neural activity Kriegeskorte & Douglas (2019). Decoding models have been explored across various neuroimaging modalities, including intracranial EEG (iEEG) (Chakrabarti et al., 2015; Akbari et al., 2019; Wang et al., 2023), functional magnetic resonance imaging (fMRI) (Naselaris et al., 2011; Tang et al., 2023), magnetoencephalography (MEG) (Défossez et al., 2022; Wang et al., 2024), and electroencephalography (EEG) (Wang & Ji, 2022; Liu et al., 2024). These models have been applied to diverse settings such as imagined and perceived language (Wang & Ji, 2022; Défossez et al., 2022; Tang et al., 2023), speech reconstruction (Akbari et al., 2019; Li et al., 2024), motor control (Robinson & Vinod, 2016; Pandarinath et al., 2017), and vision (Nishimoto et al., 2011; Zou et al., 2023; Xia et al., 2024; Benchetrit et al., 2024). Of particular note are recent efforts showcasing the ability of these models to decode motor intention for speech at near real-time speeds with high accuracy (Willett et al., 2023; Metzger et al., 2023). Such models have the potential to revolutionize speech therapies for those who suffer from maladies that make it difficult to produce speech, such as locked-in syndrome (Birbaumer, 2006; Luo et al., 2023). However, these approaches primarily focus on decoding motor intentions, which may not capture the full richness of linguistic semantic content.

An alternative to decoding motor intention of speech is decoding the semantics of intended speech from elsewhere in cortex (Huth et al., 2016; Rybář & Daly, 2022). While semantic decoding has been investigated using fMRI and MEG (Tang et al., 2023; Dash et al., 2020), there is a less research leveraging the higher temporal resolution and signal quality of iEEG for this purpose (Makin et al., 2020). Despite the potential advantages of using iEEG for semantic decoding, existing methods face significant challenges when adapting to this domain, particularly due to data scarcity. To address these limitations, we propose Neuro2Semantic, a novel framework that employs transfer learning to efficiently decode the semantics of perceived speech from iEEG recordings with limited data availability.

Our approach has two main parts. First, we train an LSTM (Hochreiter & Schmidhuber, 1997) adapter to align neural data with a pre-trained text embedding space (Raffel et al., 2020; OpenAI,

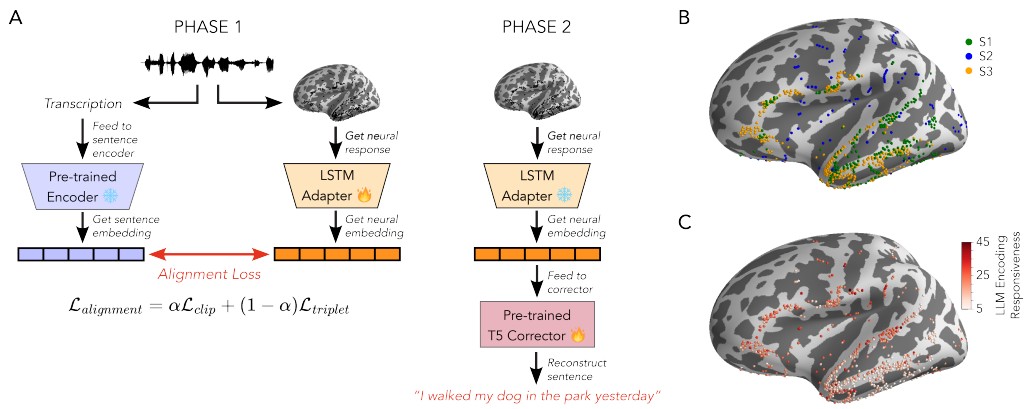

Figure 1: Neuro2Semantic architecture and training methodology. A) Training is split into 2 phases. B) Electrode coverage by subject. C) Only electrodes with significant encoding of LLM embeddings were used. Responsiveness was measured using a $t$-test comparing the encoding strength of Mistral-7B layer 15 embeddings. Electrodes are colored by $-\ln p$ using the p-value from an independent $t$-test between shuffled and non-shuffled embeddings over 10 splits.

2022) using a contrastive loss function. Second, after aligning the neural embeddings, we fine-tune a pre-trained text reconstruction model (Morris et al., 2023) to extract coherent text from the neural-aligned embeddings. This step allows for unconstrained text generation, moving beyond classification-based methods that are restricted to predefined vocabularies or limited sets of candidates.

We demonstrate that our Neuro2Semantic framework can successfully perform few-shot reconstruction of the meaning of perceived speech in in-domain settings with as little as 30 minutes of neural data. Moreover, it achieves strong performance in zero-shot reconstruction in out-of-domain contexts, showcasing its ability to generalize to entirely new semantic content. Our results show significant improvements over an existing state-of-the-art method, highlighting the effectiveness of transfer learning in both scenarios. This advancement opens new possibilities for developing more flexible and data-efficient neural decoding models, with potential applications in augmentative and alternative communication technologies.

## 2 METHODS

### 2.1 NEURO2SEMANTIC OVERVIEW

The proposed Neuro2Semantic framework, illustrated in Figure 1, is designed to map neural signals to their corresponding semantic content through a two-phase training process. In the first phase, an LSTM adapter processes the neural data and aligns it with text embeddings obtained from a pre-trained text embedding model (text-embedding-ada-002 (OpenAI, 2022)). This alignment is enforced using a combination of contrastive and triplet loss, ensuring that the neural representations are semantically consistent with the corresponding text embeddings by bringing matching pairs closer in the embedding space while pushing non-matching pairs apart. In the second phase, we utilize the methodology outlined in the Vec2text framework (Morris et al., 2023), which translates these text embeddings back into natural language. During this phase, the LSTM adapter is frozen, and we fine-tune the Vec2text inversion model to reconstruct the text from the aligned neural embeddings, effectively decoding the semantic information from the subjects' brain activity.

### 2.1.1 LSTM ADAPTER AND ALIGNMENT

Neuro2Semantic employs an LSTM adapter to encode iEEG signals into fixed-dimensional embeddings matching the text embedding dimensionality $d$. Leveraging LSTM's ability to capture

long-range dependencies, the adapter transforms preprocessed iEEG signals into embeddings that encapsulate relevant features, facilitating effective alignment with semantic embeddings.

To achieve effective alignment between the neural embeddings generated by the LSTM adapter and the corresponding semantic embeddings, we employ an alignment loss that combines a contrastive objective with a batch-level similarity optimization (Radford et al., 2021). The alignment loss is designed to ensure that the neural embeddings are not only semantically aligned with the text embeddings but also robustly distinguishable from non-corresponding pairs.

Let $\mathbf{E}_n = \{\mathbf{e}_n^i\}_{i=1}^N$ represent the set of neural embeddings produced by the LSTM adapter, where each $\mathbf{e}_n^i \in \mathbb{R}^d$ is an embedding corresponding to the $i$-th iEEG input. Similarly, let $\mathbf{E}_t = \{\mathbf{e}_t^i\}_{i=1}^N$ denote the set of corresponding text embeddings, where each $\mathbf{e}_t^i \in \mathbb{R}^d$ represents the semantic embedding associated with the $i$-th input.

First, we normalize the embeddings to ensure they lie on a unit hypersphere:

$$\hat{\mathbf{e}}_n^i = \frac{\mathbf{e}_n^i}{\|\mathbf{e}_n^i\|_2}, \quad \hat{\mathbf{e}}_t^i = \frac{\mathbf{e}_t^i}{\|\mathbf{e}_t^i\|_2} \tag{1}$$

The similarity matrix $\mathbf{S}$ between the normalized neural and text embeddings is then computed as:

$$\mathbf{S}_{ij} = \frac{\hat{\mathbf{e}}_t^i \cdot \hat{\mathbf{e}}_n^j}{\tau} \tag{2}$$

where $\tau$ is a temperature parameter that controls the sharpness of the similarity distribution. The diagonal elements $\mathbf{S}_{ii}$ represent the similarity between corresponding pairs, while off-diagonal elements $\mathbf{S}_{ij}$ (for $i \neq j$) capture the similarity between non-corresponding pairs.

The alignment objective includes two components:

**CLIP-based Contrastive Loss:**

$$\mathcal{L}_{\text{clip}} = \frac{1}{2N} \left[ -\sum_{i=1}^N \log \frac{\exp(\mathbf{S}_{ii})}{\sum_{j=1}^N \exp(\mathbf{S}_{ij})} - \sum_{i=1}^N \log \frac{\exp(\mathbf{S}_{ii})}{\sum_{j=1}^N \exp(\mathbf{S}_{ji})} \right] \tag{3}$$

This loss operates on the cosine similarities between the neural and text embeddings, ensuring that each neural embedding is most similar to its corresponding text embedding in the joint embedding space.

**Triplet Margin Loss:**

Given an anchor embedding $\mathbf{e}_n^i$, the corresponding positive embedding $\mathbf{e}_t^i$, and a randomly chosen negative embedding $\mathbf{e}_t^k$ (where $k \neq i$), the triplet margin loss is defined as:

$$\mathcal{L}_{\text{triplet}} = \sum_{i=1}^N \max\left(0, \|\mathbf{e}_n^i - \mathbf{e}_t^i\|_2^2 - \|\mathbf{e}_n^i - \mathbf{e}_t^k\|_2^2 + \delta\right) \tag{4}$$

This loss minimizes the mean squared error (MSE) between corresponding pairs of neural and text embeddings, while enforcing a margin $\delta$ that separates these pairs from non-matching ones.

The final alignment loss is a weighted combination of these two objectives:

$$\mathcal{L}_{\text{alignment}} = \alpha \mathcal{L}_{\text{clip}} + (1 - \alpha) \mathcal{L}_{\text{triplet}} \tag{5}$$

where $\alpha$ controls the trade-off between the CLIP loss and the triplet margin loss, ensuring that the neural embeddings are both closely aligned with their corresponding text embeddings and sufficiently distinct from non-corresponding pairs.

### 2.1.2 VEC2TEXT CORRECTOR MODULE

The Neuro2Semantic framework transforms aligned neural embeddings into coherent text using the Vec2Text method (Morris et al., 2023). Vec2Text, pre-trained on large-scale text corpora, learns a robust mapping from dense text embeddings to discrete text sequences. It frames the inversion task as a controlled generation problem, aiming to generate text $x$ whose embedding $\hat{\mathbf{e}}(x)$ closely approximates the target embedding $\mathbf{e}$.

The model operates iteratively, starting with an initial hypothesis $x^{(0)}$ and refining it over multiple steps $t$. At each step, the model minimizes the distance between the current hypothesis embedding $\hat{\mathbf{e}}(x^{(t)})$ and the target embedding $\mathbf{e}$, progressively enhancing the coherence and accuracy of the generated text. Mathematically, the goal is to solve the following optimization problem:

$$\hat{x} = \arg \max_x \cos(\hat{\mathbf{e}}(x), \mathbf{e}) \tag{6}$$

Here, $\cos(\hat{\mathbf{e}}(x), \mathbf{e})$ represents the cosine similarity between the embedding of the generated text and the target embedding. The optimization seeks to find the text sequence $x$ that maximizes this similarity.

The iterative refinement process can be expressed as:

$$x^{(t+1)} = \arg \max_x p(x|\mathbf{e}, x^{(t)}, \hat{\mathbf{e}}(x^{(t)})) \tag{7}$$

where $p(x|\mathbf{e}, x^{(t)}, \hat{\mathbf{e}}(x^{(t)}))$ is the probability distribution over possible next texts conditioned on the target embedding $\mathbf{e}$, the current hypothesis $x^{(t)}$, and its corresponding embedding $\hat{\mathbf{e}}(x^{(t)})$. The model updates the text hypothesis by comparing the embedding of the current hypothesis $\hat{\mathbf{e}}(x^{(t)})$ with the target embedding $\mathbf{e}$, and generating a new text hypothesis that is more aligned with $\mathbf{e}$.

The Vec2Text model employs an encoder-decoder transformer architecture conditioned on the previous text hypothesis $x^{(t)}$ and the target embedding $\mathbf{e}$. This iterative process continues until the cosine similarity $\cos(\hat{\mathbf{e}}(x), \mathbf{e})$ converges, resulting in text $x$ that accurately reflects the original semantic and syntactic structure of the text.

### 2.1.3 FINE-TUNING THE CORRECTOR MODULE

The second phase of the Neuro2Semantic framework focuses on transforming the aligned neural embeddings into coherent text sequences. This is accomplished by fine-tuning the Vec2Text corrector module (Morris et al., 2023), which is designed to invert text embeddings back into their original textual form. Although the Vec2Text model is pre-trained on large-scale text corpora, fine-tuning it with neural embeddings allows the model to adapt to the specific characteristics of neural embeddings, enhancing its ability to accurately reconstruct the original text from these embeddings.

During fine-tuning, the LSTM adapter is kept frozen to preserve the semantic alignment established in the first phase. Only the parameters of the Vec2Text corrector module are updated. The process begins by passing the preprocessed iEEG segments through the LSTM adapter to generate fixed-dimensional neural embeddings $\mathbf{e}_n$. These embeddings, now aligned with the text embedding space, serve as input to the Vec2Text corrector, which aims to reconstruct the original text sequences $x = (x_1, x_2, \ldots, x_T)$.

The training objective is to maximize the likelihood of generating the correct text sequence given the neural embeddings. We employ the negative log-likelihood (NLL) loss, a standard choice for sequence-to-sequence models. The loss function is defined as:

$$\mathcal{L}_{\text{NLL}} = -\sum_{t=1}^{T} \log p(x_t|\mathbf{e}_n, x_{<t}) \tag{8}$$

where $p(x_t|\mathbf{e}_n, x_{<t})$ is the probability of predicting the token $x_t$ at time step $t$, conditioned on the neural embeddings $\mathbf{e}_n$ and the previous tokens $x_{<t} = (x_1, x_2, \ldots, x_{t-1})$.

## 2.2 Intracranial Recordings and Data Processing

Three subjects undergoing surgical evaluation for drug-resistant epilepsy participated. Stereotactic EEG electrodes were implanted intracranially (iEEG) for epileptogenic localization. Any electrodes showing signs of epileptiform discharges, as identified by an epileptologist, were removed from analysis. Prior to electrode implantation, all subjects provided written informed consent for research participation. The subjects listened to naturalistic recordings of people engaging in podcast-like conversations between several speakers. There were six different conversations used, each discussing a different topic or situation, and each was further split into several subsections (trials) with an average duration of approximately one minute. In total, the task included about 30 minutes of speech. These trials enabled a quick check for the subject's attention by asking the subject a question about what was just said. All subjects were able to answer the questions at the end of each trial without difficulty. The research protocol was approved by the governing institutional review board.

The envelope of the high-gamma band ($70 - 150$ Hz) of the neural recordings during listening was computed using the Hilbert transform (Edwards et al., 2009) and downsampled to $100$ Hz Mischler et al. (2023). The high-gamma band was used due to its correlation with neuronal firing rates (Ray & Maunsell, 2011; Steinschneider et al., 2008). We restricted our method to use only the electrodes which were significantly responsive to semantic features. To do this, we followed the methodology of (Mischler et al., 2024) to predict electrode responses from the layer $15$ embeddings of the large language model Mistral-7B (Jiang et al., 2023) and then analyzed only the electrodes with significant predictions scores compared to the scores achieved when shuffling the Mistral-7B embeddings between words ($p < 0.05$) (Fig. 1C). To overcome the limited electrode coverage inherent in iEEG recordings, we combined the electrodes from all three subjects into a single subject for all analyses, totalling $864$ electrodes.

## 2.3 Baseline Model

For a baseline comparison, we use the Bayesian decoding method proposed by Tang et al. (2023) to generate decoded stimuli. In short, the method uses a beam search to generate proposed continuations to a beam of candidate decoding texts. Encoding models generated per the methodology of Mischler et al. (2024) are then used to evaluate the proposed continuations and rank them based on the likelihood that they correspond to the observed neural responses. As in the original work, we model the likelihood $p(R|S)$ of observing brain responses $R$ given a stimulus $S$ using a multivariate Gaussian distribution with mean $\mu = \hat{R}(S)$ and covariance $\Sigma$, where $\hat{R}(S)$ is the predicted neural response for stimulus $S$, and $\Sigma$ is estimated as the covariance of residuals from the encoding model over the training data. Slight modifications have been made to adjust the baseline model to work with iEEG data, in particular, we use encoding models generated using the high-gamma band of the neural recordings, and we use fewer and shorter finite impulse response (FIR) delays to account for the lack of a delayed hemodynamic response curve. We selected this method as a point of comparison because it represents recent state-of-the-art results in fMRI decoding and is the most closely related to our approach in terms of both objective—perceived speech semantic reconstruction—and methodology—generative-based decoding

# 3 Experiments and Results

## 3.1 Experimental Setup

**Training Procedure**. We trained the model using a leave-one-out cross-validation approach, where the last trial of each story was left out for testing. Each trial was split into sentences, with the corresponding neural data segment from when the sentence was spoken used for training. This setup prevented any anti-causal leakage of information when fine-tuning the language models while allowing the model to train on the semantic content of past sentences within the same conversation. This process was repeated for each of the six stories, with model performance evaluated after each epoch using cross-validation. The held-out trial from each story served as the test set for that split.

During Phase 1, the LSTM adapter was trained for 100 epochs with a batch size of 8, using the Adam optimizer (Kingma & Ba, 2015) with a learning rate of 1.3e-3. Once the adapter training was

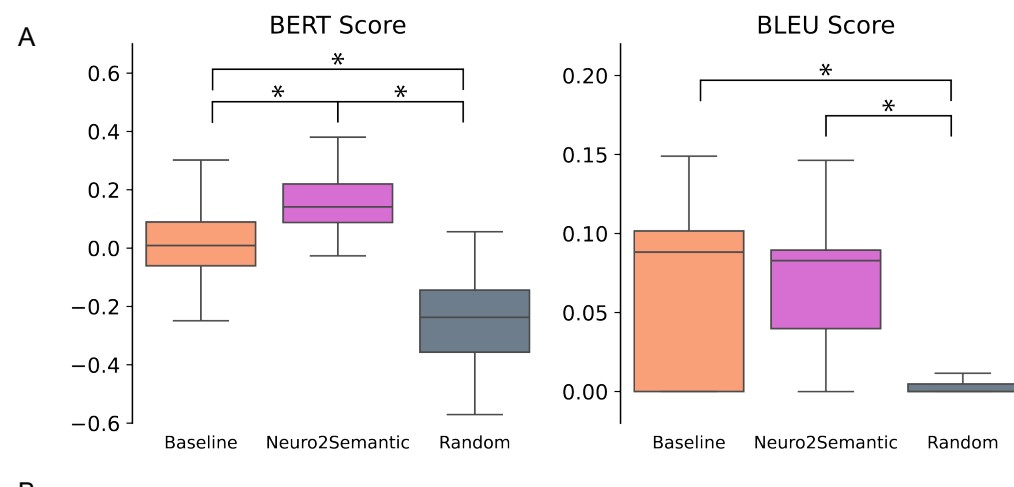

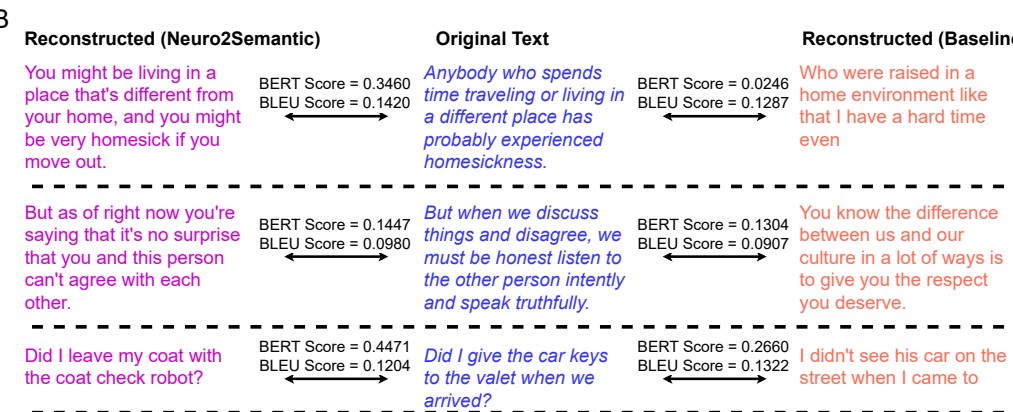

Figure 2: Performance comparison between Neuro2Semantic, baseline, and random control (A) Boxplots of BERT Score (left) and BLEU Score (right) comparing the performance of Neuro2Semantic, the baseline model (Tang et al., 2023), and a random control. Significance is indicated with * (p < 0.05) based on a paired *t*-test. B) Sample handpicked sentence reconstructions from Neuro2Semantic (left), the baseline model (right), and the original text (middle).

complete, its parameters were frozen for Phase 2, where the pre-trained corrector was fine-tuned for 2 epochs. In this phase, the corrector used only one step for the refinement process.

The CLIP-based contrastive loss was employed with a temperature parameter of $\tau = 0.1$, and the $\alpha = 0.25$ term was used to control the contribution between the contrastive loss and triplet margin loss. Ablation studies on the effect of these hyperparameters on the performance of the model are provided in A.1. All training was conducted on a single NVIDIA L40 GPU.

**Evaluation Metrics.** To evaluate the quality of the reconstructed text, we used several metrics commonly applied in text generation tasks, including BLEU (Papineni et al., 2002), Word Error Rate (WER), ROUGE (Lin, 2004), and BERT Score (Zhang et al., 2020). These metrics provided insights into both the surface-level and semantic accuracy of the generated text compared to the original. Further details on how these metrics were used and calculated are provided in A.2.

### 3.2 PERFORMANCE COMPARISON: NEURO2SEMANTIC, BASELINE, AND PHASES OF TRAINING

We evaluate Neuro2Semantic against a baseline model from previous work (Tang et al., 2023) and a random control to rigorously assess our approach's effectiveness. To ensure a balanced evaluation, results in Table 1 are averaged across the left-out trial from each of the six stories, ensuring equal contribution from each story regardless of the number of sentences per trial. Figure 2A presents box-

Table 1: Performance comparison between Neuro2Semantic, the baseline model (Tang et al., 2023), and training phases of Neuro2Semantic. Metrics are reported as mean ± standard deviation. Significant improvements over random are marked with * (p < 0.05) based on a paired *t*-test.

| Model | BERT ↑ | BLEU ↑ | ROUGE ↑ | WER ↓ |
|---|---|---|---|---|
| Random | -0.2452 ± 0.1328 | 0.0026 ± 0.0034 | 0.0321 ± 0.0471 | 1.00 ± 0.00 |
| Baseline (Tang et al., 2023) | 0.0315 ± 0.1273 * | 0.0642 ± 0.0536 * | 0.1131 ± 0.1205 * | 0.9746 ± 0.0632 * |
| **Neuro2Semantic (Full Model)** | **0.1947 ± 0.1283** * | **0.0789 ± 0.0627** * | **0.1387 ± 0.1159** * | **0.9660 ± 0.0774** * |
| Neuro2Semantic - Adapter Only (Phase 1) | 0.0560 ± 0.0864 * | 0.0677 ± 0.0386 * | 0.0841 ± 0.0726 * | 0.9943 ± 0.0232 |
| Neuro2Semantic - Corrector Only (Phase 2) | 0.1001 ± 0.0992 * | 0.0346 ± 0.0450 * | 0.0679 ± 0.0785 * | 0.9816 ± 0.0387 * |

plots illustrating the distribution of performance metrics across all sentence pairs, providing a comprehensive view of each model's variability and consistency. Additionally, Figure 2B demonstrates sample reconstructed sentences from each model alongside the original ground truth transcriptions, highlighting the qualitative improvements achieved by Neuro2Semantic.

Our Neuro2Semantic model significantly outperforms the baseline, particularly in BERT Score, indicating its suitability for low-data settings. Additionally, it achieves higher scores in BLEU and ROUGE, reflecting improved 4-gram overlap and recall-based measures, respectively. The random control, which generates sentences using randomly initialized LSTM adapter and corrector modules, consistently underperforms across all metrics.

To further understand the contributions of each phase in Neuro2Semantic, we analyze the performance impact of the two training stages: Phase 1 (LSTM adapter) and Phase 2 (Vec2Text corrector). Phase 1 involves aligning neural data with pre-trained text embeddings using contrastive loss, while Phase 2 fine-tunes the corrector module to map these aligned embeddings back into text. As shown in Table 1, the full Neuro2Semantic model, integrating both phases, achieves the highest performance across all metrics (BERT Score: 0.1947, BLEU: 0.0789, ROUGE: 0.1387, WER: 0.9660). Phase 1 alone (LSTM adapter) yields moderate results (BERT Score: 0.0560, BLEU: 0.0677), underscoring the importance of neural-text alignment in capturing semantic information. Conversely, Phase 2 alone (Vec2Text corrector) underperforms (BERT Score: 0.1001, BLEU: 0.0346), demonstrating that the alignment established in Phase 1 is critical for accurate semantic decoding. This indicates that the combined effects of both phases are essential for achieving optimal performance.

### 3.3 ZERO-SHOT OUT-OF-DOMAIN PERFORMANCE

While the previous evaluations assessed Neuro2Semantic's performance in settings where the model encountered familiar semantic contexts, it is also important to evaluate how well the model generalizes to completely unseen semantics. In this section, we explore the zero-shot out-of-domain performance by holding out entire stories that the model has not been exposed to during training. This provides a more challenging test of the model's robustness and its ability to reconstruct coherent semantic content when faced with new contexts.

Figure 3A presents a bar plot showing Neuro2Semantic's scores compared to the baseline model for BERT, BLEU, ROUGE, and 1 - WER metrics. Neuro2Semantic consistently outperforms the baseline across all metrics. In particular, the BERT score shows a substantial improvement, suggesting that the model can maintain semantic coherence even when exposed to entirely new stories. BLEU and ROUGE scores indicate moderate improvements in n-gram overlap and recall-based measures, respectively. The WER has been inverted (1 - WER) to align with the other metrics, showing that Neuro2Semantic achieves lower error rates than the baseline. Error bars highlight the variance across different stories.

In addition to these quantitative results, Figure 3B shows example sentences reconstructed by Neuro2Semantic for out-of-domain stories. For Story 3, which poses more challenges, the model generates a reconstruction ("I'm looking at some TV shows about how people could really live in a modern place") that, while not exact, still captures aspects of the original semantic content ("This is the place with the robotic waiters, right?"). The mention of a "modern place" shows that the model picked up on the notion of a unique or futuristic environment, somewhat paralleling "robotic wait-

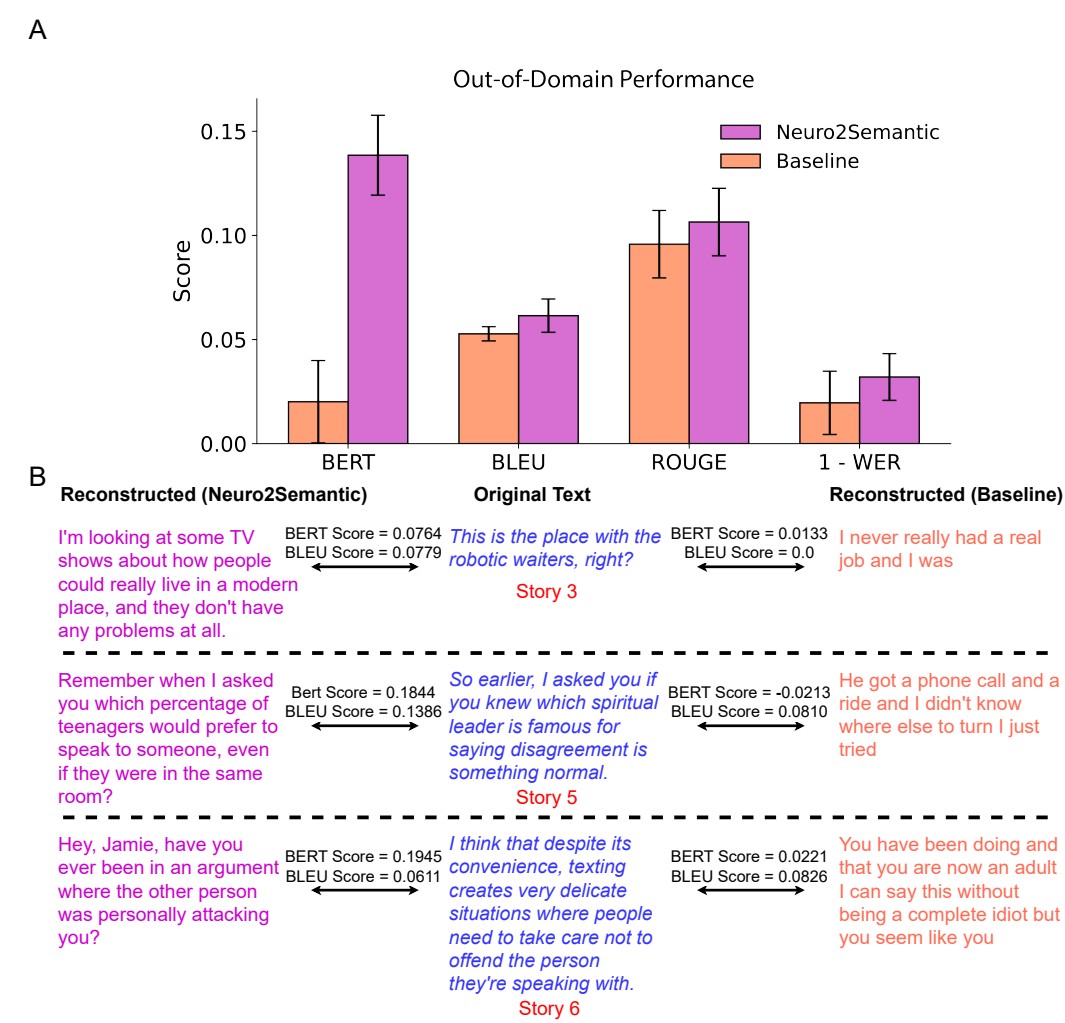

Figure 3: (A) Zero-shot out-of-domain performance comparison for Neuro2Semantic and baseline across 6 stories, using BERT, BLEU, ROUGE, and 1 - WER metrics. Higher values indicate better performance across all metrics. (B) Sample handpicked sentence reconstructions from Neuro2Semantic (left), original text (center), and the baseline model (right).

ers." Meanwhile, Similarly, Story 5 retains the conversational nature of the original, even though the content diverges, showing the model's ability to maintain structure and flow. For Story 6, the reconstruction captures the essence of conflict and interaction from the original text, emphasizing Neuro2Semantic's strength in extracting underlying themes from neural data. In contrast, the baseline model outputs less coherent and relevant sentences, further highlighting Neuro2Semantic's superior generalization in out-of-domain settings.

## 3.4 IMPACT OF DATA AND ELECTRODE SCALING ON MODEL PERFORMANCE

In this section, we evaluate how scaling both the training data and the number of electrodes affects the performance of the Neuro2Semantic model. First, we assess the impact of scaling the training data by training the model on random subsets of 20%, 40%, 60%, 80%, and 100% of the available data. For each subset percentage, five independent runs were conducted, with the standard deviation across runs displayed as error bars in Figure 4A. As the training data increases, we observe significant performance improvements that appear almost linear across all evaluation metrics, including BERT Score, BLEU, ROUGE, and WER. This demonstrates that larger datasets enhance

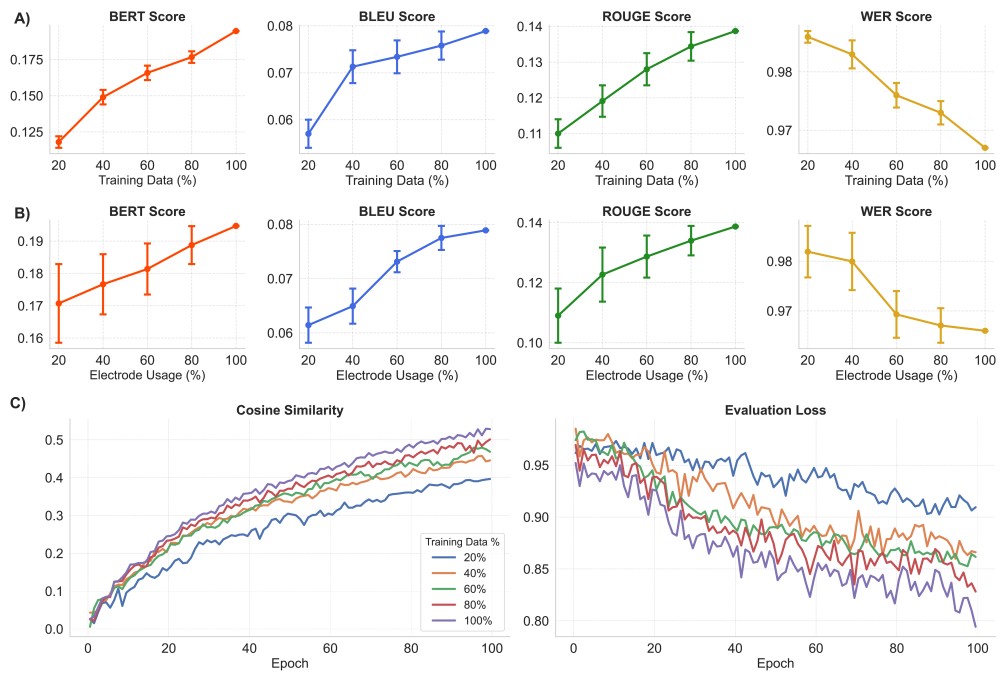

Figure 4: Performance of Neuro2Semantic across different percentages of training data (A) and electrode usage (B). Metrics include BERT, BLEU, ROUGE, and WER scores, with error bars representing the standard deviation across different runs. (C) Cosine similarity between neural and text embeddings and evaluation loss across training epochs for various percentages of training data.

the model's ability to generalize, leading to more accurate text reconstruction. This emphasizes the model's potential can be significantly boosted with access to even larger datasets

Similarly, we investigated the effect of varying electrode usage by training the model on a random subset of 20%, 40%, 60%, 80%, and 100% of the available electrodes. We ran the experiment five times with a different selected subset for each percentage. The results are presented in Figure 4B. The relatively large error bars illustrate that in some runs, selecting 80% of the electrodes can achieve similar accuracy to using 100% of the electrodes. This is a crucial observation, as it suggests that Neuro2Semantic is capable of maintaining high accuracy with fewer electrodes. The ability to achieve comparable performance with reduced electrode coverage highlights the data and resource efficiency of the model, making it particularly valuable in practical settings where dense electrode sampling may be challenging or impractical to achieve.

In Figure 4C, we show the cosine similarity between the evaluation neural and text embeddings, along with the evaluation loss over epochs for different training data percentages. Larger datasets lead to faster convergence and higher cosine similarity, indicating better alignment between neural and text representations as training data scales.

## 4 DISCUSSION

### 4.1 COMPARISON TO PREVIOUS WORK

To rigorously assess the effectiveness of Neuro2Semantic, we compared it against the baseline method presented by Tang et al. (2023). This baseline was chosen due to its close alignment with our task and objective—decoding perceived speech into continuous, semantically rich text from neural signals. In contrast to Wang & Ji (2022), which focuses on reading tasks using a sequence-to-sequence machine translation framework, and Défossez et al. (2022), which employs retrieval-based decoding with acoustic features from a pre-trained Wav2Vec2 model (Baevski et al., 2020), our ap-

proach emphasizes direct semantic reconstruction from neural embeddings. Additionally, Makin et al. (2020) utilizes classification-based techniques with predefined vocabularies, thereby limiting flexibility in text generation. Given these distinctions, the methodology of Tang et al. (2023), which achieves state-of-the-art performance in semantic decoding and continuous text generation using fMRI data, serves as the most relevant and appropriate baseline for our study.

**Efficient Data Utilization Through Transfer Learning**. By leveraging transfer learning, our method utilizes pre-trained text embeddings, allowing the model to effectively decode text with significantly less neural data—just 30 minutes compared to the much larger datasets used in prior work, such as 16 hours of fMRI data in Tang et al. (2023), 26 hours of MEG data in Défossez et al. (2022), and 6 hours of EEG data in Wang & Ji (2022). Through our two-phase training process, Neuro2Semantic achieves high performance while requiring a fraction of the training data. When replicating Tang et al. (2023)'s method on our dataset, Neuro2Semantic outperformed it by nearly six times higher BERT scores, underscoring the efficacy of our model. Additionally, Neuro2Semantic shows promising scalability, highlighting the potential for significant performance gains with expanded datasets.

**Zero-Shot Generalization.** One of the strengths of Neuro2Semantic is its ability to generalize in zero-shot settings, successfully reconstructing text from previously unseen semantic contexts. This distinguishes it from models like Tang et al. (2023), which are limited by their reliance on specific training sets and vocabularies. Neuro2Semantic's zero-shot capabilities demonstrate its robustness and potential for application in more flexible and expansive brain-to-text decoding tasks.

**Generative Text Decoding for Flexible Brain-to-Text Translation.** Our approach stands out for its generative text decoding capability, which directly reconstructs continuous, natural language from neural embeddings. Unlike some previous methods that rely on classification-based or retrieval-based approaches (Makin et al., 2020; Défossez et al., 2022), our model enables flexible text generation without requiring predefined vocabularies or word rate estimation, as in Tang et al. (2023). This flexibility allows for a broader range of possible outputs, making it more adaptable to real-world scenarios where semantic decoding is required.

## 4.2 LIMITATIONS

**High Dependence on Pre-trained Text Embeddings and Vec2Text Corrector**. The model's performance is bottlenecked by how accurately the Vec2Text corrector (Morris et al., 2023) can reconstruct text from embeddings. While pre-trained text embeddings capture much of the semantic content, the corrector's ability to translate those embeddings back into coherent text is crucial. This dependency introduces challenges, particularly when dealing with domain-specific or rare vocabulary, where the pre-trained embeddings might not fully capture the necessary semantic nuances, limiting the effectiveness of the text reconstruction. Additionally, bias in the distribution of semantics that deviates from the regular distribution of natural language in the training data can overtune the corrector to sometimes hallucinate phrases and terms that appear frequently in the training data.

**Cross-subject Variability**. While the model shows strong performance with only 30 minutes of data, its generalizability may be impacted by variability in neural recordings across subjects or different brain regions. To fully validate its robustness, more extensive testing on larger, more diverse datasets involving multiple subjects and brain regions is necessary.

## 4.3 CONCLUSION

We introduced Neuro2Semantic, a novel framework for decoding continuous language from neural signals, leveraging transfer learning to map neural activity to text. By using pre-trained text embeddings and a two-phase training process, our model achieves strong performance with just 30 minutes of data, significantly outperforming previous methods like Tang et al. (2023). Neuro2Semantic excels in unconstrained text generation and zero-shot generalization, highlighting its potential for real-world applications in brain-computer interfaces and natural language processing. Additionally, the framework scales effectively with larger datasets and can operate well with limited brain coverage, making it a practical and efficient choice for neurotechnology applications. This advancement marks a significant step forward in brain-to-text decoding, opening up impactful opportunities in communication and assistive technologies.

## 5 ETHICS STATEMENT

In conducting this research, we strictly adhered to ethical guidelines to protect both the rights of participants and the integrity of our scientific process. Neural data collection was conducted with full informed consent from participants, who were thoroughly briefed on the nature and purpose of the study, as well as their rights to withdraw at any time. All data was anonymized to protect participant confidentiality, and the study was approved by the relevant institutional review board (IRB). Upon publication, we will make the IRB details available for full transparency.

We also recognize the potential risks and ethical concerns associated with brain-computer interface technology, particularly regarding mental privacy and the possibility of misuse in non-consensual surveillance or manipulation. Our commitment to ethical research includes advocating for safeguards and responsible use of neural decoding technology. We encourage ongoing ethical oversight and dialogue as the field advances to ensure that this technology is used for the benefit of society, with strict measures to prevent any harmful or unethical applications.

## 6 REPRODUCIBILITY STATEMENT

To ensure the reproducibility of our work, we have made extensive efforts to provide clear and detailed descriptions of our methodology, model architecture, and experimental setup throughout the main paper. The core components of the Neuro2Semantic framework, including the pre-trained text embeddings, LSTM adapter, and Vec2Text corrector, are fully detailed in the Methods section, with all hyperparameters, loss functions, and training configurations outlined. Although we are unable to share the dataset publicly due to patient privacy concerns, we will provide anonymous access to the source code used in our experiments as part of the supplementary materials. We will also release our trained model's checkpoints upon publication. The data processing steps for the neural signals and text embeddings are thoroughly described in A.6, allowing researchers to replicate our methods using their own datasets.

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

## A APPENDIX

### A.1 ABLATION ON HYPERPARAMETERS: $\alpha$ AND $\tau$

The parameter $\alpha$ in the Neuro2Semantic framework controls the balance between the contrastive loss and the triplet margin loss during Phase 1 of training. Adjusting $\alpha$ allows us to fine-tune the influence of these two loss components, impacting the neural-text alignment and overall performance of the model. In this section, we investigate how varying $\alpha$ affects the model's ability to reconstruct meaningful text by examining performance across BERT, BLEU, ROUGE, and WER metrics. Understanding the optimal weighting between the contrastive and triplet losses is crucial for improving the model's semantic decoding capabilities.

Table 2: Performance of Neuro2Semantic across different values of $\alpha$. Metrics include BERT, BLEU, ROUGE, and WER.

| $\alpha$ | BERT ↑ | BLEU ↑ | ROUGE ↑ | WER ↓ |
|---|---|---|---|---|
| 0.0 | 0.1508 ± 0.1047 | 0.0696 ± 0.0628 | 0.1052 ± 0.0914 | 0.9771 ± 0.0542 |
| 0.25 | **0.1947 ± 0.1283** | **0.0789 ± 0.0627** | **0.1387 ± 0.1159** | **0.9660 ± 0.0774** |
| 0.5 | 0.1643 ± 0.1277 | 0.707 ± 0.0791 | 0.1072 ± 0.1137 | 0.9702 ± 0.0853 |
| 0.75 | 0.1405 ± 0.1367 | 0.0674 ± 0.0578 | 0.1026 ± 0.1261 | 0.9839 ± 0.1033 |
| 1.0 | 0.1284 ± 0.1370 | 0.0647 ± 0.0449 | 0.0920 ± 0.0888 | 0.9876 ± 0.0394 |

Table 2 presents the results for different $\alpha$ values, showing that an intermediate $\alpha = 0.25$ yields the best overall performance across all metrics, with a BERT score of 0.1947 and a BLEU score of 0.0789. Increasing or decreasing $\alpha$ beyond this optimal value leads to a decline in performance, highlighting the importance of balance between the two loss components.

The parameter $\tau$ represents the temperature in the CLIP-based contrastive loss function. The temperature $\tau$ controls the sharpness of the similarity distribution between neural embeddings and corresponding text embeddings. A lower $\tau$ value produces a more fine-grained, focused similarity distribution, making the model more sensitive to small differences between embeddings, while a higher $\tau$ smoothens the distribution.

Table 3 shows that $\tau = 0.1$ yields the best overall performance, with a BERT score of 0.1947 and a BLEU score of 0.0789. Increasing $\tau$ beyond this leads to a decline in performance, with metrics like BERT and BLEU gradually dropping as $\tau$ rises to 0.3, indicating reduced semantic and n-gram alignment. Interestingly, at $\tau = 0.05$, BLEU scores peak at 0.08348, but this comes with a slightly lower BERT score, suggesting improved surface-level accuracy but reduced overall semantic consistency.

Table 3: Performance of Neuro2Semantic across different values of $\tau$. Metrics include BERT, BLEU, ROUGE, and WER.

| $\tau$ | BERT ↑ | BLEU ↑ | ROUGE ↑ | WER ↓ |
|---|---|---|---|---|
| 0.05 | 0.1797 ± 0.1184 | **0.08348 ± 0.0630** | 0.1384 ± 0.1120 | 0.9672 ± 0.0667 |
| 0.1 | **0.1947 ± 0.1283** | 0.0789 ± 0.0627 | **0.1387 ± 0.1159** | **0.9660 ± 0.0774** |
| 0.2 | 0.1883 ± 0.1352 | 0.0752 ± 0.0675 | 0.1304 ± 0.1183 | 0.9711 ± 0.0810 |
| 0.3 | 0.1691 ± 0.1414 | 0.0730 ± 0.0719 | 0.1203 ± 0.1225 | 0.9750 ± 0.0864 |

### A.2 EVALUATION METRICS

**BERT Score**: BERT Score was used to assess the semantic similarity between the reconstructed text and the original, leveraging pre-trained transformer models to calculate token-level cosine similarity between embeddings. BERT Score provides a more nuanced evaluation by comparing the meaning of the text rather than just surface-level token matches. We utilized the 'bert-score' library with rescaling based on English baselines. We utilized the `bert-score` library with rescaling based on

English baselines. For evaluation, we focused on the *F1 score*. The F1 scores were averaged across all test samples to obtain the final results.

**BLEU Score**: The BLEU (Bilingual Evaluation Understudy) score was used to measure n-gram overlap between the reconstructed and original texts. Given the relatively short sentence lengths in our dataset, we applied smoothing (using method 7 from NLTK's (Bird et al., 2009) 'Smoothing-Function') to avoid penalizing the model excessively for missing rare words. BLEU was computed at the sentence level, and the overall score was averaged across all sentences in the test set. BLEU measures how closely the reconstructed text matches the original in terms of shared n-grams.

$$\text{BLEU} = \exp\left(\min\left(1 - \frac{r}{c}, 0\right)\right) \prod_{n=1}^{N} (p_n)^{w_n} \tag{9}$$

where $r$ is the reference length, $c$ is the candidate length, and $p_n$ is the precision of n-grams. Weights $w_n$ were uniformly distributed across the n-grams for $n = 1, 2, 3, 4$.

**Word Error Rate (WER)**: WER was used to measure the token-level accuracy of the reconstructed text. It calculates the ratio of the number of insertions, deletions, and substitutions needed to transform the reconstructed text into the reference text, normalized by the total number of words in the reference text. A lower WER indicates fewer errors and a more accurate reconstruction.

$$\text{WER} = \frac{S + D + I}{N} \tag{10}$$

where $S$ is the number of substitutions, $D$ is the number of deletions, $I$ is the number of insertions, and $N$ is the total number of words in the reference.

**ROUGE Score**: We employed ROUGE-1 to evaluate the unigram overlap between the reconstructed and original texts. ROUGE-1 measures the number of overlapping unigrams, emphasizing how well the model preserves individual word matches. This metric focuses on recall, which highlights how much of the original text content is retained in the reconstructed version.

For our evaluation, we computed ROUGE-1 scores at the sentence level, using the F1 score. All calculations were performed using the `python-rouge` package (Lin, 2004). The ROUGE-1 score is calculated by dividing the number of matching unigrams between the reference and reconstructed text by the total number of unigrams in the reference, providing insight into how closely the reconstructed text aligns with the original.

### A.3 IMPACT OF THE TEXT EMBEDDING MODELS

In the Neuro2Semantic framework, the choice of text embedding model plays a critical role in aligning neural data with semantic content. Different embedding models capture varying levels of linguistic and contextual information, which can significantly affect the model's ability to decode accurate and meaningful text from neural signals. In this section, we explore how the performance of Neuro2Semantic varies when using different pre-trained text embedding models, specifically comparing OpenAI's text-embedding-ada-002 and Google's GTR-base language models (Ni et al., 2022). We evaluate their impact on semantic reconstruction by comparing metrics such as BERT, BLEU, ROUGE, and WER to identify which model provides better alignment and improved text reconstruction.

Table 4: Comparison of Neuro2Semantic Performance with Different Text Embedding Models. The table compares text-embedding-ada-002 and GTR-base text embedding models across BERT, BLEU, ROUGE, and WER metrics.

| Embedding Model | BERT ↑ | BLEU ↑ | ROUGE ↑ | WER ↓ |
|---|---|---|---|---|
| text-embedding-ada-002 | **0.1947 ± 0.1283** | **0.0789 ± 0.0627** | **0.1387 ± 0.1159** | 0.9660 ± 0.0774 |
| GTR-base | 0.1743 ± 0.1150 | 0.0734 ± 0.0591 | 0.1299 ± 0.1103 | **0.9502 ± 0.0689** |

Table 4 compares the performance of Neuro2Semantic using OpenAI's text-embedding-ada-002 and Google's GTR-base embedding models. While both models deliver competitive results, text-embedding-ada-002 demonstrates slightly better performance in terms of BERT, BLEU, and ROUGE, indicating stronger semantic and n-gram alignment. However, GTR-base achieves a marginally lower WER, suggesting a slight advantage in word-level accuracy. These results highlight the trade-offs between the two embeddings

### A.4 Case Studies of Decoded Text from the Brain

#### A.4.1 Interesting Cases

**Original Text:** And Rob, what did you say?
**Reconstructed Text:** What about you, Eliza?

**BERT Score:** 0.3058

The model demonstrates a solid understanding of the conversational structure here, capturing the intent behind the question. While the name is substituted, it maintains the flow and intent of an inquiry directed at a person in a dialogue. This shows the model's ability to generate plausible conversational turns, suggesting that it has internalized patterns of human dialogue and can adapt them fluidly to the context.

**Original Text:** So, Georgina, what should you do if a friend texts me when I'm already speaking to someone else in person?
**Reconstructed Text:** And someone might be speaking to you in person, but it would be ruder for them than to stop that conversation and speak to the person over text.

**BERT Score:** 0.1902

This example is a strong demonstration of the model's ability to maintain the overall conversational dynamics and meaning. It successfully grasps the social context of balancing in-person communication with texting, which involves complex social cues. The model even adds nuance with phrases like "it would be ruder," showcasing its ability to understand and convey opinions about social interactions—an impressive feat for text generated from neural signals.

**Original Text:** Do you think that explanation makes sense?
**Reconstructed Text:** What do you agree with?

**BERT Score:** 0.3203

This example highlights the model's ability to preserve the conversational nature of the original text while slightly shifting the focus of the question. Instead of directly asking about understanding, the reconstructed text refocuses on agreement, which is still within the realm of inquiry but shifts the nuance. This shows the model's flexibility in rephrasing questions, and adapting the meaning while maintaining a similar conversational flow. The relatively high BERT score reflects the semantic similarity between the original and reconstructed texts.

**Original Text:** I see.
**Reconstructed Text:** I guess.

**BERT Score:** 0.3658

This is a subtle but interesting reconstruction. Both phrases are used in casual conversations to express understanding or acknowledgment. The model's ability to produce a semantically equivalent phrase shows that it can handle short, idiomatic expressions well.

**Original Text:** As divers descend deep below the ocean's surface, there is an increase in atmospheric pressure.
**Reconstructed Text:** It seems to be too expensive, and I think the whole idea is to connect the sea surface and the deep sea to create a vertical connection.

**BERT Score:** 0.2915

In this case, the model captures the concept of a relationship between the ocean's surface and its depths, aligning with the original theme of diving and pressure. While the reconstructed text intro-

duces a new context related to cost and infrastructure ("too expensive" and "create a vertical connection"), the idea of connecting different layers of the ocean shows that the model retains a general understanding of depth and surface relationships. The performance demonstrates the model's ability to grasp abstract concepts even if the specific details diverge. This is a compelling example of how the model can still maintain thematic relevance despite producing novel content.

**Original Text:** If we don't challenge things and search for truth and justice, things would always stay the same and never develop.
**Reconstructed Text:** And if you want to disagree, the truth can be very important.

**BERT Score:** 0.2310

The model shows a clear understanding of the theme of "truth" present in the original text. While the reconstruction simplifies the original idea, it still emphasizes the importance of truth in decision-making and discussion, demonstrating that the model captures key concepts even when it rephrases more complex statements.

A.4.2 MODEL FAILURE ANALYSIS

Despite the overall success of the Neuro2Semantic model, there are several cases where the model struggles to reconstruct text accurately, particularly in maintaining the semantic meaning of short or context-specific phrases:

**Original Text:** Alright, Robot Number Nineteen, here is the ticket.
**Reconstructed Text:** Of course, there's a story about twelve Norse gods having a dinner party and one god whose name is Loki?

**BERT Score:** -0.0015

In this example, the model significantly diverges from the original meaning, substituting 'Robot Number Nineteen' with 'Twelve Norse gods' and introducing an unrelated narrative about mythology. While this represents a clear failure in capturing the correct context, it is notable that the model retained and substituted a numerical element ('Nineteen' with 'Twelve'). This suggests that, even in failure, the model's internal mechanisms are capable of recognizing numeric patterns, albeit misaligned with the expected content.

**Original Text:** That record was set by Roger Garcia, an ex-military diver and head of the world's only underwater research station.
**Reconstructed Text:** That's how we've heard of it.

**BERT Score:** 0.0606

Here, the model fails to retain any of the specific names and facts that it has not seen during the training. Instead of reconstructing "Roger Garcia" and his role, the model produces a vague response, losing all crucial information. This failure indicates the model's struggle with retaining proper nouns and complex attributions in neural alignment. Named entities tend to be challenging, as they may not be well represented in the embeddings or training data.

**Original Text:** Physiological reactions like the bends are caused by divers incorrectly readjusting to normal atmospheric pressure.
**Reconstructed Text:** With the huge rise of human activity, the phrase "oh, shut up" has become more and more popular, and it's thought that people are trying to have a conversation with the land.

**BERT Score:** 0.0800

In the original text, the subject is clearly scientific and technical, referring to the physiological effects of divers incorrectly adjusting to pressure ("the bends"). The language is precise, referencing a specific medical condition tied to diving. However, the reconstructed text diverges completely, introducing a vague social commentary about human activity and communication with "the land," which is both abstract and entirely irrelevant to the original context. This failure exemplifies how the model can lose the context of highly specific technical information, shifting to more colloquial or abstract content when faced with domain-specific terminology. It highlights a limitation in handling scientific language and suggests a need for improved embedding alignment when dealing with specialized topics like diving and physiology. Despite the failure, the sentence still retains a somewhat

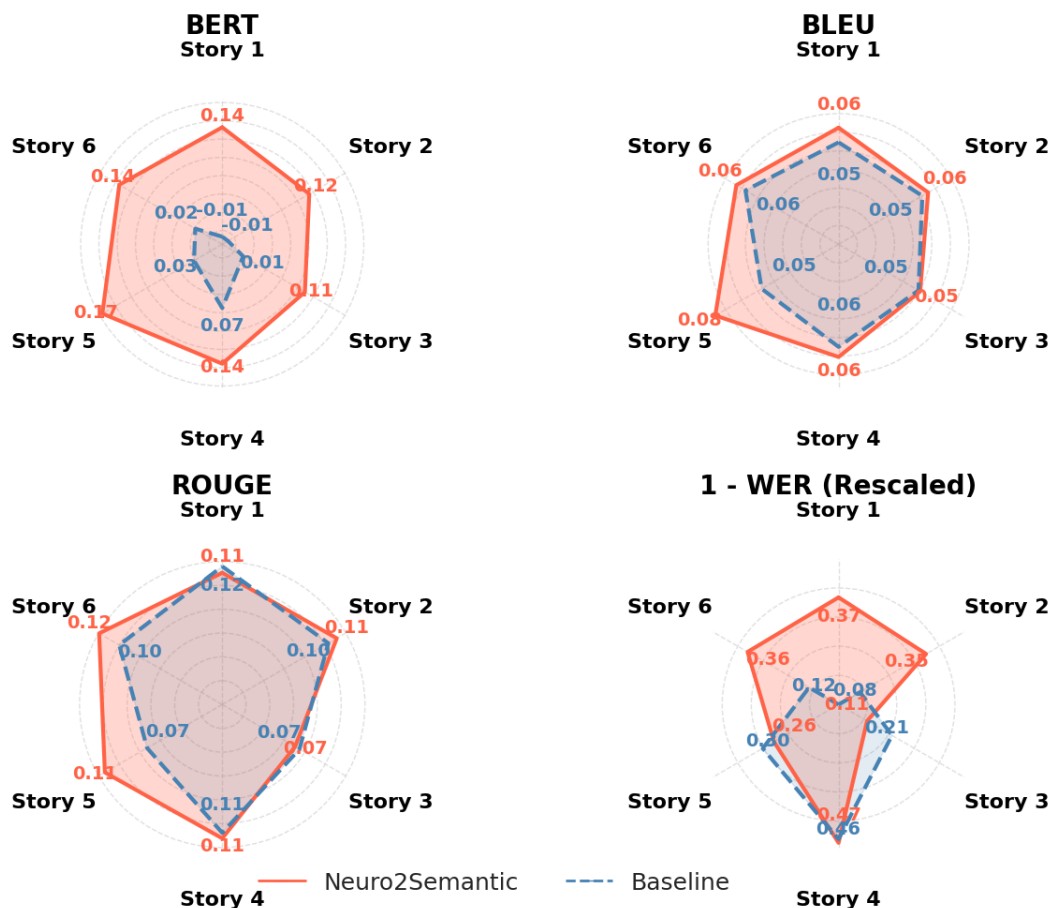

Figure 5: Zero-shot out-of-domain performance comparison Neuro2Semantic vs Baseline for each left-out story

structured format, showing the model's ability to generate complete sentences, albeit with incorrect semantic content.

### A.5   ZERO-SHOT OUT-OF-DOMAIN PER STORY

Figure 5 presents the zero-shot out-of-domain performance comparison between Neuro2Semantic and the Baseline model across six stories. Each radar plot shows the metrics—BERT, BLEU, ROUGE, and rescaled WER—highlighting the performance of both models for each story. Neuro2Semantic consistently outperforms the Baseline, particularly in BERT and ROUGE scores, suggesting that it better captures the semantic content of the stories. The plots also reveal variations in performance across different stories, indicating that certain narratives may pose more challenges for both models. Overall, this comparison underscores the effectiveness of Neuro2Semantic in reconstructing meaningful text from neural data in unfamiliar contexts.

### A.6   FURTHER METHODOLOGY FOR PREPROCESSING NEURAL RESPONSES

Neural signals were acquired at 1024 Hz sampling rate and the envelope of the high-gamma band signal ($70 - 150$ Hz) was extracted. This was done using a filter bank of gaussian filters and averaging the outputs of the filters, as in (Edwards et al., 2009), using the `filter_hilbert` function in the `naplib-python` package Mischler et al. (2023). This high-gamma envelope was then downsampled to 100 Hz. The neural data was recorded simultaneously with the stimuli, and word alignments were extracted using `prosodylab-aligner` Gorman et al. (2011), a widely used

forced aligner based on a Hidden Markov Model. From these word alignments, neural data could be extracted around words or sentences for input into the models.

## A.7 DEMONSTRATION OF THE EFFECT OF TRAINING ON NEURAL AND TEXT EMBEDDINGS

In this section, we visually demonstrate how training affects the alignment of neural and text embeddings. Throughout the training process, we observe that the neural embeddings produced by our model move closer to the corresponding text embeddings, highlighting the successful learning of semantic relationships between the brain activity and the textual representations. This alignment improves as training progresses

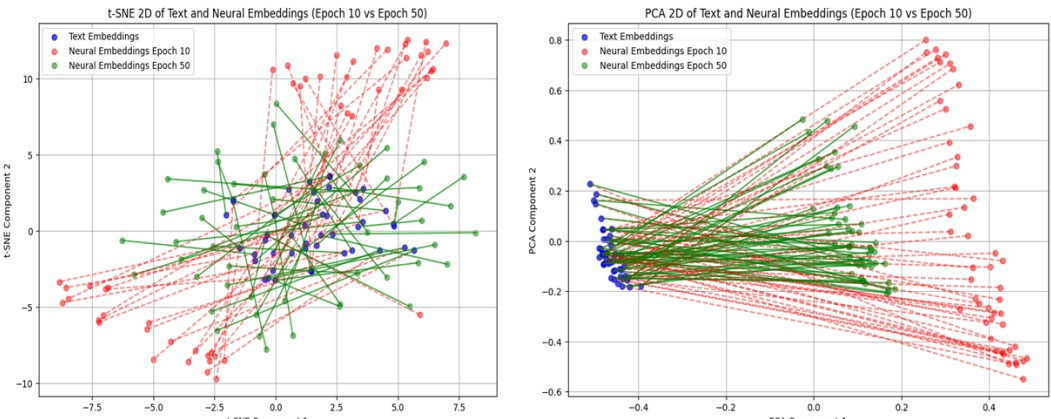

Figure 6: t-SNE visualization of text embeddings (blue) and neural embeddings from epoch 10 (red) and epoch 50 (green). The arrows show how the neural embeddings progressively move towards the corresponding text embeddings during training, demonstrating better alignment over time.



Figure 7: Cosine similarity heatmaps showing the alignment between text and neural embeddings at different stages of training: initial (Epoch 0), intermediate (Epoch 10), and final (Epoch 100). The diagonal becoming more prominent indicates that the model learns to closely align corresponding pairs while keeping non-corresponding embeddings dissimilar.

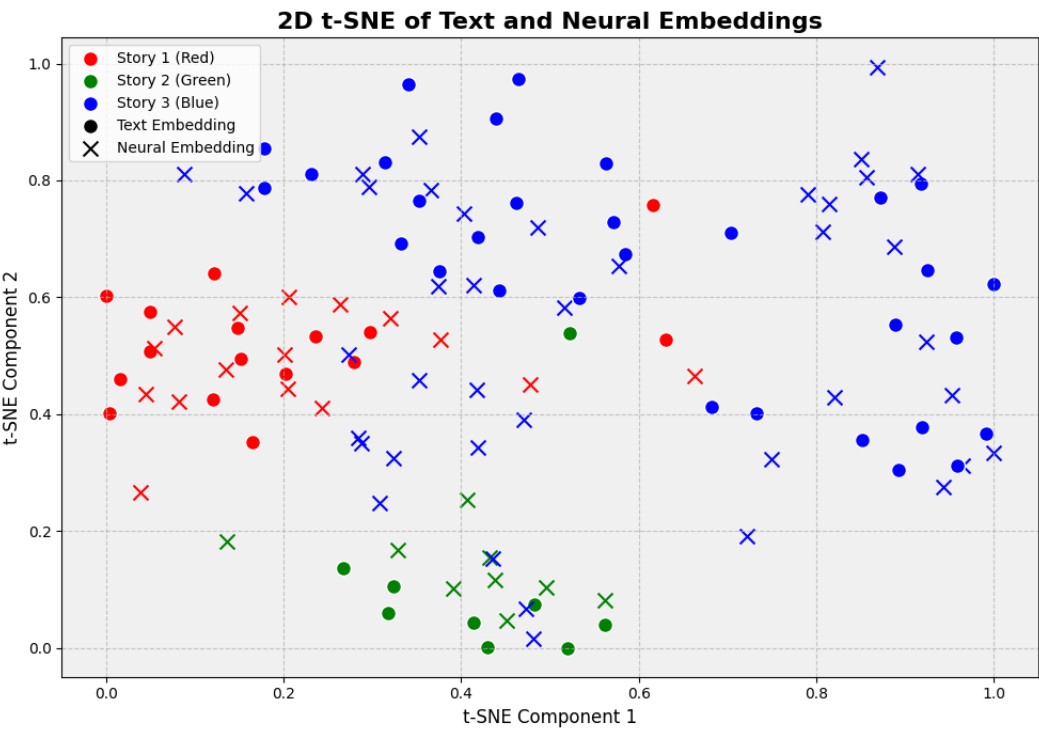

Figure 8: t-SNE plot of neural and text embeddings categorized by three stories (Story 1: red, Story 2: green, Story 3: blue). Text embeddings are represented by circles, and neural embeddings are marked by crosses. This shows the relationship between the neural embeddings and the text embeddings, with distinct clustering patterns.

