# OpenReview forum: "Neuro2Semantic: A Transfer Learning Framework for Semantic Reconstruction of Continuous Language from Human Intracranial EEG"
_ICLR.cc/2025/Conference — ICLR 2025 Conference Withdrawn Submission_

### Official Review · Reviewer_v5EM · 2024-10-24

**Soundness:** 3
**Presentation:** 3
**Contribution:** 3
**Rating:** 3
**Confidence:** 4

**Summary:**

Neuro2Semantic introduces a novel approach for reconstructing semantic content from intracranial EEG (iEEG) recordings. The authors propose a two-phase framework: In the first phase, an LSTM adapter aligns neural signals with pre-trained text embeddings using a combination of contrastive and triplet loss functions. In the second phase, a pre-trained Vec2text corrector generates continuous text from the aligned embeddings. The model demonstrates the ability to decode coherent semantic content with as little as 30 minutes of neural data, significantly outperforming baseline methods in low-data scenarios.

**Strengths:**

The paper presents a novel approach by using the Vec2text corrector module to reconstruct sentences directly from neural embeddings, which is a fresh and innovative method in the field of iEEG-based neural decoding. The combination of pre-trained text embeddings and a correction module provides new insights and directions for generating coherent text from brain signals, offering a new perspective compared to traditional classification-based or retrieval-based approaches.

**Weaknesses:**

The paper demonstrates several strengths, but there are some concerns regarding the choice of methodology, data robustness, and presentation of results. Specifically, the use of LSTM instead of more modern architectures like Transformer, the potential for overfitting with such limited data, and the incomplete dataset description raise questions about the generalizability and scalability of the approach. For detailed concerns and clarifications, please refer to the Questions section.

**Questions:**

1） The authors have chosen an LSTM-based approach for aligning neural signals with text embeddings. While LSTMs are known for handling long-range dependencies, the current state-of-the-art models for EEG and language tasks often rely on Transformer architectures, which are more powerful in capturing contextual information. Why did the authors opt for LSTM instead of using a Transformer encoder or even more advanced architectures like xLSTM? Wouldn't these architectures better align with the pre-trained Transformer-based language models used in the later stages of the pipeline?


2） The authors used the Vec2text corrector module to reconstruct sentences from neural embeddings, so it seems that the main goal would be to make the neural embeddings as similar as possible to the text embeddings. Given this, why was a more complex loss function—combining contrastive and triplet loss—chosen over a simpler Mean Squared Error (MSE) loss? What performance advantages does this combined loss offer for aligning the neural embeddings with the text embeddings?


3）This paper achieves remarkable performance with as little as 30 minutes of neural data. It raises concerns about the robustness and overfitting of the model when applied to different subjects, which suggests that more rigorous validation methods are needed to assess the model's robustness and generalizability. Implementing more comprehensive validation strategies, such as leave-one-subject-out or similar cross-validation techniques, would provide stronger evidence of the model’s ability to generalize across diverse subjects and conditions.


4）The authors propose the use of a pre-trained T5 Corrector to reconstruct sentences from text embeddings. However, autoregressive language models, which generate text one token at a time, are the standard in most generative tasks. What are the advantages of using the Corrector compared to autoregressive generation? How does it impact the model’s performance in terms of coherence and quality?


5）The presentation of decoding results in Figure 2(B) could be clearer. There is no discussion of whether important named entities or key phrases were successfully decoded. The authors could improve the clarity by bolding successfully decoded words or phrases and italicizing words with similar meanings. This would help highlight the strengths and weaknesses of the model's semantic reconstruction capabilities.


6）The dataset description lacks important details. Key demographic information, such as the subjects' age, gender, and other relevant characteristics, is missing, which could affect the generalizability of the results. Additionally, the content of the audio stimuli is not clearly explained, making it difficult to assess the consistency or variability in what the subjects heard. More information about the stimuli is necessary to fully understand the experimental setup and ensure the results are reliable.

---

> ### Author Response · Authors · 2024-11-21
>
> We sincerely appreciate your thoughtful review and the valuable feedback you've provided on our manuscript. Below, we address your concerns in detail:
>
> **[Q1]**: *"The authors have chosen an LSTM-based approach for aligning neural signals with text embeddings... "*
>
> *Response:* We chose an LSTM-based approach for aligning neural signals with text embeddings because our dataset is limited in size. Transformers require significantly more data to avoid overfitting and achieve robust performance. Empirically, we found that a simple 2-layer LSTM provided the best trade-off between performance and generalization.
>
> **[Q2]**: *"The authors used the Vec2text corrector module to reconstruct sentences from neural embeddings ..."*
>
> *Response:* Thank you for asking this question. A simple MSE loss does not distinguish between matching and non-matching neural and text pairs, treating all distances equally. In our task, it is crucial that each neural segment is as close as possible to its corresponding decoded text while remaining dissimilar to other texts. The combination of contrastive and triplet losses ensures this by enforcing both global alignment (contrastive loss) and local margin constraints (triplet loss).
>
> **[Q3]**: *"This paper achieves remarkable performance with as little as 30 minutes of neural data. It raises concerns about the robustness and overfitting of the model when applied to different subjects, which suggests that more rigorous validation methods ... "*
>
> *Response:* We acknowledge your concern about overfitting. We combined all subjects primarily to enhance brain coverage in our experiments. Additionally, our out-of-domain experiments demonstrate the model's ability to generalize to unseen semantic contexts, providing some evidence of its robustness. However, we agree that leave-one-subject-out or similar cross-validation methods would provide a more rigorous evaluation of generalizability, and we plan to explore these approaches in future work as we collect more data for this task.
>
> **[Q4]**: *"The authors propose the use of a pre-trained T5 Corrector to reconstruct sentences from text embeddings..."*
>
> *Response:* Thank you for raising this question. To clarify, the Vec2Text Corrector module in our approach operates auto-regressively, generating text one token at a time, similar to standard generative models. Its primary distinction lies in its iterative refinement mechanism. This approach complements traditional auto-regressive generation by enabling fine-grained corrections, which enhances alignment with the underlying embeddings.
>
> **[Q5]**: *"The presentation of decoding results in Figure 2(B) could be clearer. There is no discussion of whether important named entities or key phrases were successfully decoded...."*
>
> *Response:* We appreciate your suggestion to improve the clarity of decoding results in Figure 2(B) by emphasizing key phrases and named entities. We will incorporate this feedback in the camera-ready version by bolding successfully decoded words and italicizing words with similar meanings to better highlight the model's semantic reconstruction strengths and weaknesses.
>
>
> **[Q6]**: *"The dataset description lacks important details. Key demographic information, such as the subjects' age, gender, and other relevant characteristics, is missing..."*
>
> *Response:* We appreciate your thoughtful suggestion. We acknowledge that including detailed demographic information about the subjects and a clearer description of the audio stimuli would provide a more comprehensive understanding of the experimental setup. We will incorporate these details in the camera-ready version.

---

### Official Review · Reviewer_o8Aw · 2024-10-28

**Soundness:** 3
**Presentation:** 3
**Contribution:** 2
**Rating:** 5
**Confidence:** 3

**Summary:**

The submission presents an approach for decoding continuous perceived sentences using intracranial EEG recordings. This method comprises two stages: the first stage involves aligning neural signals with text in a latent space, and the second stage utilizes a Vec2Text module to map the generated latent embeddings to text. The results demonstrate that the proposed method outperforms baseline approaches in most of the cases regarding both in-domain and out-of-domain scenarios.

**Strengths:**

The submission is well written and clear. Authors conducted comprehensive experiments. The results demonstrates the effectivness of the methods outperforms the baseline method the scenario for in domain and out of domain cases.

**Weaknesses:**

Generally, the semantic meaning of the reconstructed sentences in both Neuro2Semantic and the baseline method appears to diverge significantly from the original text, as illustrated in Figures 2B and 3B. While the word error rate (WER) captures only surface-level matching between the decoded sentences and the ground truth, it is crucial to retain certain key nouns to ensure accurate semantic transmission. Current word error rate is very high, also other metrics like BERT score or BLEU score is quite low. The authors could enhance their discussion by elaborating on how the current method contributes to the field, particularly given the limited semantic alignment between the decoded sentences and the ground truth, despite improvements in the metrics.

Additionally, it would be beneficial to include a related work section that compares the proposed method with previous approaches focused on decoding continuous sentences from neural signals. The authors could also discuss related works on methods that learn the alignment of two modalities, as well as those that generate continuous vectors from text embeddings.

**Questions:**

1. Did authors try different architectures for phase one, instead of using LSTM?
2. How many sentences/trials are used evaluated to obtain current results?
4. What could be the upper bound of the performance, especially when more data is provided?
5. Why not include model from Wang et al., 2024 as baseline comparison?
6. What is the intuition behind using both Triplet loss and the contrastive loss since they are serving the same function?
7. It is interesting that, using phase 2 alone, without any alignment between text and neural signal, the model can still generate some outputs with comparable BLUE score to the full model, could author describe more on this? Also how the random control baseline is performed?

---

> ### Author Response · Authors · 2024-11-21
>
> We sincerely appreciate your thoughtful review and the valuable feedback you've provided on our manuscript. Below, we address your concerns in detail:
>
> **[W1]**: *"Generally, the semantic meaning of the reconstructed sentences in both Neuro2Semantic and the baseline method appears to diverge significantly from the original text"*
>
> *Response:* Thank you for your comment. Semantic decoding from neural data, regardless of modality (e.g., EEG, iEEG, MEG, fMRI), remains a challenging task, that no existing methods have fully resolved. Our work demonstrates substantial progress in this area by leveraging transfer learning to achieve results comparable to or better than existing methods in similar conditions.
>
> **[W2]**: *"it would be beneficial to include a related work section that compares the proposed method with previous approaches focused on decoding continuous sentences from neural signals"*
>
> *Response:* We appreciate this valuable suggestion. While our current manuscript discusses related works relevant to our methodology, we acknowledge that further expanding the discussion to include studies focused on the alignment of modalities and related advancements would strengthen our presentation. We appreciate this suggestion and will expand our related work section in the camera-ready version.
>
> **[Q1]**: *"Did authors try different architectures for phase one, instead of using LSTM?
> "*
>
> *Response:* Yes, we have experimented with both Transformers and GRUs. Transformers performed worse than LSTMs due to the limited size of our dataset, as they require more data to effectively capture long-term dependencies without overfitting. GRUs did not offer substantial improvements, so we opted for a 2-layer LSTM to keep the aligning model simple and better suited to our data constraints.
>
>
> **[Q2]**: *"How many sentences/trials are used evaluated to obtain current results?
> "*
>
> *Response:* Our dataset consisted of 30 trials and 419 unique sentences. We used the last trial of each story for evaluation (6 trials = 89 sentences)
>
>
> **[Q3]**: *"What could be the upper bound of the performance, especially when more data is provided?
> "*
>
> *Response:* While it's challenging to estimate a precise upper bound, our scalability experiments (Section 4.4) suggest that performance improves significantly with more training data, as larger datasets enable better alignment between neural and text embeddings. The point you raised suggests a promising path for future research.
>
>
> **[Q4]**: *"Why not include model from Wang et al., 2024 as baseline comparison?
> "*
>
> *Response:* The approach used in (Wang et al., 2024), is identical to that of (Tang et al, 2023)--our current baseline.
>
> **[Q5]**: *"What is the intuition behind using both Triplet loss and the contrastive loss since they are serving the same function?
> "*
>
> *Response:* While both Triplet and Contrastive losses aim to align neural embeddings with text embeddings, they complement each other by optimizing different aspects of the alignment. Contrastive loss ensures global separation of matching and non-matching pairs, while Triplet loss enforces a margin between positive and negative pairs, improving local embedding consistency. Empirically as  we showed in the appendix combination of the two results in better alignment.
>
> **[Q6]**: *"It is interesting that, using phase 2 alone, without any alignment between text and neural signal, the model can still generate some outputs with comparable BLUE score to the full model, could author describe more on this? Also how the random control baseline is performed?
> "*
>
> *Response:* While the BLEU score for Phase 2 alone is approximately half of the full model, it is relatively high because fine-tuning the corrector on the training stories allows it to adapt stylistically, leading to text that resembles the story context regardless of the neural embeddings. However, Phase 1 is critical for aligning neural signals with text embeddings, ensuring the decoded sentences correspond to the actual neural data. For the random control baseline, we used untrained models in both phases, which produced irrelevant text, confirming that both alignment and fine-tuning are necessary for meaningful semantic decoding.

---

> > ### Comment · Reviewer_o8Aw · 2024-11-24
> >
> > Regarding Q5, To me contrastive loss and triplet loss are the same thing, especially based on your description here "enforces a margin between positive and negative pairs" will results in "global separation of matching and non-matching pairs", while I assume "positive" = "matching", "negative"="non-matching". Of course, they might be slightly different based on your implementation. But in appendix, the performance when $\alpha=0$ is very close to $\alpha=0.25$.
> >
> > Regarding Q2, it seems the dataset is very small containing only 419 unique sentences. It is very likely the method is overfitted to current dataset. How the hyperparameters are selected for the proposed method vs. for the baseline models.
> >
> > Regarding Q6, it would be interesting to see what the evaluation metrics if random/noise signals are sent to the fully trained model.

---

### Official Review · Reviewer_RNym · 2024-11-01

**Soundness:** 2
**Presentation:** 2
**Contribution:** 2
**Rating:** 3
**Confidence:** 4

**Summary:**

The paper presents Neuro2Semantic, a new framework for decoding the semantic content of perceived speech from intracranial EEG (iEEG) recordings. The approach has two main components: an LSTM-based adapter that maps neural signals to pre-trained text embeddings, and a corrector module that produces continuous, natural language from these embeddings.  The paper claims that it could outperforms state-of-the-art models while only requiring 30 minutes of neural recordigns.

**Strengths:**

The performance reported in the experimental section shows good improvement over baseline models. It shows the BLEU score reached 0.1947 while the baseline method only shows 0.0315.

**Weaknesses:**

1. The experiment is not adequate to claim that this is a solid SOTA for brian-to-text translation, especially only given 30 minutes of training data. Considering this method has a T5 corrector over the previous method, the well-trained language model will generate multiple semantically continuous sentences containing many articles and common words will give the illusion of an artificially high bleuscore. Figure 2 generated text samples also support this phenomenon. Meanwhile, do the training set and the test set share similar sentences? If so, the training set split could also largely affect the performance rather than really learning neural patterns.

Especially, when referring to Table 1 Neuro2Semantic - Adapter Only, which emphasizes the neural patterns. This paper reaches a BLEU score of 0.0677 which is about the same as the baseline of 0.0642 and the ROUGE score is 0.1131 (baseline) and 0.00841 (this paper). It shows no clear evidence in this paper to learn a better neural representation.

2. The experimental part lacks the cross-session and the cross-subject analysis. Only 30 minutes of data may lead to a severe over-fitting problem.

3. The novelty is very limited, utilize an adaptor and language model to do the brain-to-text translation task. The new thing is to use iEEG rather than EEG.  However, according to the main structure part, the only difference with EEG-based models is that iEEG uses an LSTM framework vs. others using a conformer structure. Meanwhile, some of the works are not properly included and discussed, such as [1,2].

[1] Li J, Song Z, Wang J, Zhang M, Zhang Z. BrainECHO: Semantic Brain Signal Decoding through Vector-Quantized Spectrogram Reconstruction for Whisper-Enhanced Text Generation.

[2] Yang Y, Duan Y, Zhang Q, Xu R, Xiong H. Decode neural signal as speech.

**Questions:**

1. What is the key difference between doing the iEEG-to-text translation rather than EEG-to-text translation?

---

> ### Author Response · Authors · 2024-11-21
>
> We sincerely appreciate your thoughtful review and the valuable feedback you've provided on our manuscript. Below, we address your concerns in detail:
>
> **[W1]** *"The experiment is not adequate to claim that this is a solid SOTA for brain-to-text translation ..."*
>
> *Response:* Thank you for pointing out these important considerations. We do not claim SOTA but aim to demonstrate the potential of our transfer learning paradigm, particularly in low-data settings like iEEG. To address concerns about the role of the language model (corrector), we have conducted experiments to ensure the performance is not solely due to its fluency. As shown in Table 1, random trial baselines (with models untrained in Phases 1 and 2) generate irrelevant sentences, and Phase 2 alone achieves only half the BERT and BLEU scores of the full model, underscoring the importance of the alignment phase. While our training and test examples in Section 3.2 included contextually similar sentences (e.g., about moving out for college), Section 3.3 demonstrates that the model generalizes well even to entirely unseen stories. This highlights the robustness and promise of our approach.
>
> **[W2]** *"The experimental part lacks the cross-session and the cross-subject analysis. Only 30 minutes of data may lead to a severe over-fitting problem."*
>
> *Response:* We agree that our study does not include cross-session or cross-subject analyses. However, to enhance the coverage of the brain in our experiments, we combined the electrode responses from multiple subjects to form a "super-subject." This approach allowed us to leverage a broader range of neural signals, which is particularly valuable in iEEG studies where electrode coverage is often sparse. While cross-subject validation remains an important direction for future work, this methodology was essential to demonstrate the potential of our framework with limited data.
>
> **[W3]** *"The novelty is very limited ..."*
>
> *Response:* We acknowledge the release of BrainECHO on October 19th, 2024, just 11 days before the ICLR deadline, and consider it concurrent work. While BrainECHO focuses on non-invasive EEG/MEG decoding via audio spectrogram alignment, our work addresses semantic text reconstruction from invasive iEEG recordings, emphasizing alignment of brain responses to semantic representations. The concurrent timing underscores the growing interest in this field and does not diminish the novelty of our transfer learning-based approach, which is tailored for low-data iEEG settings. We view these works as complementary advancements in brain-to-text decoding.
>
> **[Q1]**  *"What is the key difference between doing the iEEG-to-text translation rather than EEG-to-text translation?"*
>
> *Response*: The key difference lies in the nature of the data and its applications. iEEG signals have a higher signal-to-noise ratio and quality compared to EEG, making them more suitable for precise decoding tasks, particularly as future BCI devices like Neuralink are expected to be invasive. While iEEG data availability is limited due to its invasive nature, our contribution lies not in introducing a new modality but in leveraging a transfer learning paradigm combined with a direct text embedding-to-text approach (Vec2Text). This highlights the significant potential of transfer learning in low-data settings for advancing BCI applications.

---

### Official Review · Reviewer_LJ4X · 2024-11-03

**Soundness:** 3
**Presentation:** 2
**Contribution:** 3
**Rating:** 3
**Confidence:** 3

**Summary:**

This paper introduces a novel framework called Neuro2Semantic for reconstructing semantic content from human intracranial electroencephalogram (iEEG) recordings. This study breaks through the limitations of previous methods by leveraging contras loss and triplet loss and using a pre-trained text reconstruction model in the fine-tuning stage to extract coherent text.

**Strengths:**

1. The transfer learning framework adopted by Neuro2Semantic can achieve efficient semantic reconstruction under the condition of a small amount of data.
2. Neuro2Semantic is able to perform zero-shot reconstruction on completely unseen semantic content, showing good generalization.

**Weaknesses:**

1. This paper only compares one baseline, and this baseline is very poor.
2. Model performance is highly dependent on the quality and relevance of pre-trained text embeddings, which may limit its effectiveness when dealing with domain-specific or rare vocabularies.

**Questions:**

1. What is the reason for the poor performance of baseline, especially much worse than the reported values?
2. Have the authors explored other more efficient sequence processing models (such as Transformers or GRUs) to improve the capture and computational efficiency of long-term dependencies?

---

> ### Author Response · Authors · 2024-11-21
>
> We sincerely appreciate your thoughtful review and the valuable feedback you've provided on our manuscript. Below, we address your concerns in detail:
>
> **[W1]** *"This paper only compares one baseline, and this baseline is very poor."*
>
> *Response*: As discussed in the Discussion section of our manuscript, we compared our approach primarily with Tang et al. (2023) due to its status as a recent state-of-the-art in semantic decoding and its methodological alignment with our objectives, it's important to note that their model was trained on an extensive dataset of **17 hours of fMRI data**. In contrast, our Neuro2Semantic framework leverages transfer learning to achieve superior performance with as little as **30 minutes of iEEG data**. This demonstrates the effectiveness of our approach in low-data settings like ECoG studies, where collecting extensive neural data is inherently challenging.
>
>
>
> **[W2]** *"Model performance is highly dependent on the quality and relevance of pre-trained text embeddings ..."*
>
> *Response*: We acknowledge this limitation in our manuscript, attributing it partly to the limited size of our dataset. Nonetheless, our out-of-domain performance results show that even with these constraints, the model captures semantic content effectively, despite not always fully reconstructing syntactic structures. Additionally, our scalability experiments indicate that with larger datasets and more diverse vocabularies, the model's generalization capabilities would improve significantly, enhancing its effectiveness in domain-specific applications.
>
> **[Q1]** *"What is the reason for the poor performance of baseline, especially much worse than the reported values?"*
>
> *Response*: Thank you for raising this question. To clarify, could you specify which "reported values" you are referring to? Are you comparing the baseline's performance in our paper to the results reported in its original paper
>
> **[Q2]** *"Have the authors explored other more efficient sequence processing models (such as Transformers or GRUs) to improve the capture and computational efficiency of long-term dependencies?"*
>
> *Response:* Yes, we have experimented with both Transformers and GRUs. Transformers performed worse than LSTMs due to the limited size of our dataset, as they require more data to effectively capture long-term dependencies without overfitting. GRUs did not offer substantial improvements, so we opted for a 2-layer LSTM to keep the aligning model simple and better suited to our data constraints.

---

### Author Response · Authors · 2024-11-21

We sincerely appreciate the time and effort you have dedicated to reviewing our manuscript and providing thoughtful feedback. Below, we address your concerns in detail and outline our plans to incorporate necessary improvements in the camera-ready version. We hope that our clarifications and planned enhancements will address your points satisfactorily and contribute to a positive reassessment of our submission.

---

### Note · Authors · 2025-02-20

I have read and agree with the venue's withdrawal policy on behalf of myself and my co-authors.

---

### Meta-Review · Area_Chair_UnT7 · 2024-12-19

**Metareview:**

This paper introduces an approach for reconstructing semantic content in natural language from intracranial EEG (iEEG) recordings. The main contributions including collecting a new dataset of EEG and text from participants and developing a method based on pre-trained text embeddings, an LSTM model to generate text, and a post-corrector to improve the coherence of decoded text.

All reviewers found the paper interesting, both the data and proposed method, but with severe lacking in the evaluation of the proposed method. These issues include lack of comparison to recent baselines, lack of ablation studies of various objective functions and architectures, lack of sufficient samples to ensure experimental validity and generalization, and lack of many dataset details and discussions of related work. As a result, all reviewers gave a negative score for the paper. I am aligned with all the reviewers' judgement that while the idea is interesting and the dataset is a unique contribution, the lack of novelty of the approach and the lack of rigorous experimental validation makes this paper fall under the acceptance threshold.

**Additional Comments On Reviewer Discussion:**

The authors provided a rebuttal but did not address all the comments raised by the reviewers. None of the reviewers were willing to increase their negative scores.

---

### Decision · Program_Chairs · 2025-01-22

Reject